# Lactate Mediates High-Intensity Interval Training—Induced Promotion of Hippocampal Mitochondrial Function through the GPR81-ERK1/2 Pathway

**DOI:** 10.3390/antiox12122087

**Published:** 2023-12-07

**Authors:** Qinghui Shang, Xuepeng Bian, Lutao Zhu, Jun Liu, Min Wu, Shujie Lou

**Affiliations:** 1Key Laboratory of Exercise and Health Sciences, Shanghai University of Sport, Ministry of Education, Shanghai 200438, China; 2111516008@sus.edu.cn; 2Key Laboratory of Human Performance, Shanghai University of Sport, Shanghai 200438, China; 2011516007@sus.edu.cn (X.B.); 2111516011@sus.edu.cn (M.W.)

**Keywords:** high-intensity interval training, mitochondria, lactate, GPR81, ERK1/2

## Abstract

Mitochondrial biogenesis and fusion are essential for maintaining healthy mitochondria and ATP production. High-intensity interval training (HIIT) can enhance mitochondrial function in mouse hippocampi, but its underlying mechanism is not completely understood. Lactate generated during HIIT may mediate the beneficial effects of HIIT on neuroplasticity by activating the lactate receptor GPR81. Furthermore, growing evidence shows that lactate contributes to mitochondrial function. Given that mitochondrial function is crucial for cerebral physiological processes, the current study aimed to determine the mechanism of HIIT in hippocampal mitochondrial function. In vivo, GPR81 was knocked down in the hippocampi of mice via the injection of adeno-associated virus (AAV) vectors. The GPR81-knockdown mice were subjected to HIIT. The results demonstrated that HIIT increased mitochondria numbers, ATP production, and oxidative phosphorylation (OXPHOS) in the hippocampi of mice. In addition, HIIT induced mitochondrial biogenesis, fusion, synaptic plasticity, and ERK1/2 phosphorylation but not in GPR81-knockdown mice. In vitro, Neuro-2A cells were treated with L-lactate, a GPR81 agonist, and an ERK1/2 inhibitor. The results showed that both L-lactate and the GPR81 agonist increased mitochondrial biogenesis, fusion, ATP levels, OXPHOS, mitochondrial membrane potential, and synaptic plasticity. However, the inhibition of ERK1/2 phosphorylation blunted L-lactate or the GPR81 agonist-induced promotion of mitochondrial function and synaptic plasticity. In conclusion, our findings suggest that lactate mediates HIIT-induced promotion of mitochondrial function through the GPR81-ERK1/2 pathway.

## 1. Introduction

Most energy in neurons is used for synaptic transmission, making it important to maintain sufficient mitochondrial ATP production [1]. Numerous studies have demonstrated that abnormal mitochondria may be linked to synapse dysfunction [2,3,4]. Therefore, an improvement in mitochondrial function is important for synaptic plasticity. Emerging evidence has shown that the promotion of mitochondrial function benefits from high-intensity interval training (HIIT) [5,6,7], which is characterized by periods of high-intensity exercise combined with short rest intervals. However, the underlying mechanisms of HIIT on hippocampal mitochondrial function in mice have not been fully elucidated.

In neurons, mitochondria are enriched at presynaptic terminals at the ends of axons and postsynaptic terminals at the ends of dendrites, where bioenergetic demand is particularly high [8]. Mitochondrial biogenesis produces new mitochondria accompanied by mitochondrial fusion to maintain a healthy mitochondrial network [9]. Mitochondrial biogenesis and fusion play active roles in mitochondrial function. Mitochondrial biogenesis is a cellular process in which new and healthy mitochondria are formed [10]. Peroxisome proliferator-activated receptor γ coactivator 1α (PGC-1α) is a master regulator of mitochondrial biogenesis and interacts with transcription factors, including nuclear respiratory factors (NRF1 and NRF2) [11]. Both NRF1 and NRF2 participate in the transcription of mitochondrial transcription factor A (TFAM), which are major regulators of mitochondrial DNA transcription [12]. Mitochondrial fusion requires the coordination of three enzymes, mitofusin 1 and mitofusin 2 (MFN1 and MFN2), and promotes the fusion of the outer mitochondrial membrane (OMM), whereas optic atrophy 1 (OPA1) promotes the fusion of the inner mitochondrial membrane (IMM) [13]. HIIT has been shown to have positive effects on hippocampal mitochondrial biogenesis, fusion, and lactate levels in our previous work [14], indicating the potential roles of lactate in hippocampal mitochondrial function.

Despite the old perception that lactate is a metabolic end-product of glycolysis and causes acidosis in the body, recent studies have emphasized its beneficial effect. When blood lactate levels increase, such as during exercise, there is a net influx of lactate from blood to the brain as an energy substrate [15] or as a signaling molecule [16]. It has been shown that peak blood lactate during HIIT is 12.3 ± 3.5 mM [17]. Our previous research has shown that hippocampal lactate levels increase significantly in mice during HIIT [14]. The neurobiological roles of brain lactate as an energy substrate have now been described. However, the mechanisms of lactate as a signaling molecule in the HIIT-induced promotion of cerebral function are not completely understood. A previous study demonstrated that lactate mediates the effects of exercise on learning and memory by activating BDNF signaling [18]. Lactate can mediate HIIT-induced neurogenesis [19] and angiogenesis [16] in the brain via the lactate receptor (GPR81). In addition, lactate can enhance mitochondrial function in myotubes [20] and promote the gene expression of mitochondrial biogenesis in neuroblastoma cells [21]. Furthermore, our previous work has demonstrated that lactate promotes mitochondrial biogenesis and fusion in primary hippocampal cells [14], but the roles of GPR81 in hippocampal mitochondrial function remain unknown.

GPR81 is a transmembrane G-protein coupled receptor (GPCR) discovered in 2001 for which lactate is the only known endogenous ligand [22,23]. GPR81 is expressed mainly in adipose, brain, skeletal muscle, and liver tissues [23]. Recent studies have shown that GPR81 can activate the cAMP/PKA pathway [20,24,25] and the PLC/PKC/Akt pathway [26]. In addition, GPR81 can regulate the phosphorylation of the ERK1/2 protein kinases [27,28,29]. Studies have shown that ERK1/2 can mediate mitochondrial biogenesis, fusion, and mitochondrial function in lung epithelial cells [30], hair follicle stem cells [31], kidney tissue [32], and human retinal microcapillary endothelial cells [33]. However, whether HIIT can enhance hippocampal mitochondrial function through GPR81-ERK1/2 needs to be explored.

Given the existing evidence, we hypothesized that lactate produced as a consequence of HIIT can activate the GPR81-ERK1/2 pathway, which might further promote hippocampal mitochondrial biogenesis, fusion, and mitochondrial function. To test this hypothesis, we used both in vivo and in vitro studies to identify the roles of GPR81 and ERK1/2 in the HIIT-induced promotion of mitochondrial function.

## 2. Materials and Methods

### 2.1. Animals

The 7-week-old male C57BL/6J mice were purchased from the Nanjing Model Animal Research Center and kept in the SPF animal laboratory (temperature of 22 °C ± 2 °C under a 12/12 h light/dark cycle) with free access to water and food. The experimental design and animal use were in accordance with the guidelines set by the Institutional Animal Ethics Committee (IAEC) and approved by the Scientific Research Ethics Committee of Shanghai University of Sport (102772019DW010).

### 2.2. HIIT Paradigm and Blood Lactate Detection

Mice were subjected to HIIT with a protocol adapted from the study of Morland et al. [16]. Each HIIT session consisted of a 10 min warm-up at 10 m/min and ten bouts of exercise/recovery at intervals of 4 min. After the warm-up, the animals were made to run at 80–90% of the maximum speed for 4 min, and then allowed 2 min of active recovery at 40–50% of the maximum speed with a 10° inclination (Table 1). A maximal exercise-capacity test was performed for each mouse every other week to adjust the running speed to near the maximum it could sustain during ten consecutive intervals. Tail-tip blood was immediately collected from each mouse in each group after HIIT to measure blood lactate. Blood lactate was measured with Lactate-Scout (EKF Co., Leipzig, Germany).

### 2.3. Adeno-Associated Virus (AAV) Treatment

To study the roles of GPR81 in HIIT-mediated mitochondrial function, AAVs were bilaterally infused into the hippocampus using stereotaxic surgery (AP, −2.3 mm; ML, ±1.9 mm; DV, −2.0 mm). The shGPR81/SED and shGPR81/HIIT groups were injected with AAV harboring shRNA-targeting murine GPR81 to induce the specific knockdown of hippocampal GPR81, and the shNC/SED and shNC/HIIT groups were injected with AAV harboring control short hairpin RNA. The AAVs used above were packaged and purified by HANBIO, Shanghai, China.

### 2.4. Transmission Electron Microscopy

Mice were decapitated, and the hippocampi were quickly separated from the brains. Then, the hippocampi were dissected into 1 mm^3^ pieces and placed in 2.5% glutaraldehyde for 2 h at 4 °C. Next, the hippocampi were postfixed with 1% osmium tetroxide in 0.1 M phosphate buffer (PB) for 1 h at room temperature and dehydrated in graded alcohols. The hippocampi were embedded and polymerized for 48 h at 60 °C. Pieces of CA1 regions were selected, and ultrathin sections of 60 nm were cut on a Leica Ultracut (Leica EM UC7, Wetzlar, Germany). The sections were stained with uranyl acetate and lead citrate for 15 min. Images were obtained using a transmission electron microscope (HT7700, Hitachi, Tokyo, Japan). The number of mitochondria in the CA1 region was analyzed.

### 2.5. Neuro-2A Cell Culture

Neuro-2A cells were purchased from the National Collection of Authenticated Cell Cultures, China. The cells were grown in MEM (11090-081, Gibco, New York, NY, USA) containing 10% FBS (10099141C, Gibco), 1% GlutaMax (35050061, Gibco), 1% sodium pyruvate (11360070, Gibco), 1% NEAA (11140050, Gibco), and 1% penicillin–streptomycin solution (C0222, Beyotime, Shanghai, China) at 37 °C in 5% CO_2_.

### 2.6. Cell Treatment

Neuro-2A cells were treated with L-lactate (L7022, Sigma-Aldrich, St. Louis, MO, USA) for six hours or the GPR81 agonist 3-OBA (3-chloro-5-hydroxybenzoic acid, S5400, Selleck, Houston, TX, USA) for three hours. ERK1/2 inhibitor (SCH772984, S7101, Selleck) was applied 30 min before L-lactate and 3-OBA stimulation. L-lactate was prepared in MEM at a final concentration of 15 mM. 3-OBA was prepared as a 1000 mM stock in DMSO and used at a final concentration of 1 mM. SCH772984 was prepared as a 4 mM stock in DMSO and used at a final concentration of 2 μM. Considering the possible effects of DMSO, the DMSO group received equivalent volumes of DMSO.

### 2.7. JC-1 Staining

JC-1 staining was performed using an enhanced mitochondrial membrane potential (MMP) assay kit with JC-1 (C2003S, Beyotime) to detect the MMP. The cells were pretreated with 2 μM SCH772984 and then incubated with L-lactate or 3-OBA. The cells were washed with PBS once and incubated with 2 mL of JC-1 staining working buffer per well for 20 min at 37 °C. Then, the cells were washed twice with JC-1 staining buffer solution, and 2 mL of MEM was added. The intensities of red fluorescence and green fluorescence were observed by fluorescence microscopy. MMP was measured according to the ratio of red/green fluorescence intensity.

### 2.8. ATP Detection

ATP levels were detected using an enhanced ATP assay kit (S0027, Beyotime). Mouse hippocampi and Neuro-2A cells were lysed using ATP lysis buffer on ice and centrifuged at 12,000× *g* at 4 °C for 5 min. The supernatants were collected for assessment. Then, 100 μL of ATP working solution and 20 μL of supernatant were added to a 96-well plate and mixed quickly. ATP levels were measured with a luminometer and normalized to the protein concentration.

### 2.9. RNA Extraction and Reverse Transcription-Quantitative PCR (RT-qPCR)

Total RNA from cells and tissues was extracted using TRIzol reagent (9109, Takara, Shiga, Japan). Reverse transcription was performed using a PrimeScript™ RT reagent kit (RR047A, Takara) according to the manufacturer’s protocol. Real-time PCR was performed using TB Green^®^ *Premix Ex Taq*™ II (RR820A, Takara). The primer information for qPCR is listed in Table 2.

### 2.10. Western Blot Analysis

Mouse hippocampi and Neuro-2A cells were lysed using RIPA lysis buffer (P0013B, Beyotime) containing 1% PMSF (ST506, Beyotime) on ice and centrifuged at 14,000× *g* at 4 °C for 5 min. The supernatants were collected, and the protein concentrations of the supernatants were measured using a BCA protein assay kit (P0010, Beyotime) according to the manufacturer’s protocol. Samples were boiled in loading buffer (P0015L, Beyotime) at 95 °C for 10 min. Total protein (20 μg) was loaded onto a 10% PAGE gel (PG112/PG113, Epizyme, Shanghai, China) and transferred to a PVDF membrane. The PVDF membrane was blocked in blocking buffer (BSA and TBS-Tween 20) and incubated overnight at 4 °C with the following primary antibodies: GPR81 (SAB1300790, Sigma-Aldrich), p-ERK1/2 (4370, CST, Danvers, MA, USA), t-ERK1/2 (4695, CST), Arc (16290-1-AP, Proteintech, Wuhan, China), c-Fos (2250, CST), Egr1 (4154, CST), BDNF (28205-1-AP, Proteintech), SYN (4329, CST), PSD95 (3450, CST), PGC-1α (NBP1-04676SS, Novus, Littleton, CO, USA), NRF1 (46743, CST), NRF2 (12721, CST), TFAM (ab131607, Abcam, Cambridge, UK), OPA1 (80471, CST), MFN1 (13798-1-AP, Proteintech), MFN2 (9482, CST), and β-Tubulin (10068-1-AP, Proteintech). The membrane was then washed and incubated with a horseradish peroxidase (HRP)-labeled anti-rabbit secondary antibody (SA00001-2, Proteintech) for one hour at room temperature. The protein bands were detected using an enhanced chemiluminescence kit. The densities of the protein blots were quantified using ImageJ (Version 1.54g) and normalized to the band intensity of β-Tubulin.

### 2.11. Statistical Analysis

All data were analyzed using GraphPad Prism software version 8.0 (GraphPad Software, San Diego, CA, USA). Comparisons for multiple groups were performed using one-way analysis of variance (ANOVA) followed by the Bonferroni post-tests used to measure statistical significance. A value of *p* < 0.05 was considered to indicate statistical significance.

## 3. Results

### 3.1. Knockdown of Hippocampal GPR81 Leads to a Reduction in the Number of Mitochondria in Mice

Our previous work demonstrated that HIIT can enhance hippocampal mitochondrial fusion and biogenesis, which is mainly attributable to the signaling roles of lactate [14]. Hence, on the basis of our previous research, the present study investigated the roles of the lactate receptor GPR81 in the HIIT-induced improvement of mitochondrial function. HIIT increased GPR81 protein expression, and stereotaxically injected AAV harboring shRNA against GPR81 led to the specific knockdown of GPR81 in the hippocampi of mice (Figure 1a). Compared to shNC/SED, HIIT increased the number of mitochondria (Figure 1b,c). However, GPR81 knockdown led to a reduction in the number of mitochondria (Figure 1b,c).

### 3.2. Knockdown of Hippocampal GPR81 Reduces HIIT-Induced Promotion of OXPHOS and Increases in ATP Levels in Mice

Hippocampal OXPHOS-related gene (NDUFS8, SDHb, Uqcrc1, COX5b, and Atp5a1) expression and ATP levels were used to reflect mitochondrial function in the present study. Compared to the shNC/SED group, HIIT increased the expression of OXPHOS-related genes, including *NDUFS8* (Figure 2a), *SDHb* (Figure 2b), *Uqcrc1* (Figure 2c), *COX5b* (Figure 2d), and Atp5a1 (Figure 2e), as well as the levels of ATP (Figure 2f). However, GPR81 knockdown led to a lower expression of genes associated with OXPHOS (Figure 2a,b,d,e) and lower levels of ATP (Figure 2f). Furthermore, HIIT could no longer increase the expression of OXPHOS-related genes in mice with a knockdown of GPR81 (Figure 2a,c–e). These results demonstrate that HIIT enhances mitochondrial function in a manner mediated by GPR81.

### 3.3. Knockdown of Hippocampal GPR81 Prevents HIIT from Inducing the Expression of Genes and Proteins Expression Related to Mitochondrial Biogenesis

Exercise can increase mitochondrial biogenesis, which is crucial for mitochondrial energy production. In the present study, higher expression levels of genes and proteins associated with mitochondrial biogenesis were found in shNC/HIIT mice than in shNC/SED mice (Figure 3). However, GPR81 deficiency led to a lower expression of genes and proteins associated with mitochondrial biogenesis (Figure 3b,d–h). Furthermore, a comparison of shGPR81/HIIT mice to shGPR81/SED mice revealed that HIIT failed to promote mitochondrial biogenesis-associated gene and protein expression. These results indicate that HIIT promotes mitochondrial biogenesis through a GPR81-dependent mechanism.

### 3.4. Knockdown of Hippocampal GPR81 Impairs HIIT-Induced Expression of Genes and Proteins Related to Mitochondrial Fusion

Mitochondrial fusion benefits energy transmission in metabolically active cells [34]. In the present study, the expression of mitochondrial-fusion-related genes and proteins was increased after HIIT (Figure 4). When comparing the shGPR81/SED mice to the shNC/SED mice, the knockdown of GPR81 led to a lower expression of genes and proteins associated with mitochondrial fusion. Furthermore, a comparison of shGPR81/HIIT mice to shGPR81/SED mice revealed that HIIT failed to promote mitochondrial fusion-associated gene and protein expression. These results indicate that GPR81 can regulate HIIT-induced hippocampal mitochondrial fusion.

### 3.5. Knockdown of Hippocampal GPR81 Impairs HIIT-Induced Expression of Proteins Related to Synaptic Plasticity

Mitochondria support synaptic transmission through the production of ATP, so we further measured the expression of proteins related to synaptic plasticity. Compared with shNC/SED, HIIT increased the protein expression of synaptic plasticity components, including Arc (Figure 5b), Egr1 (Figure 5d), BDNF (Figure 5e), SYN (Figure 5f), and PSD95 (Figure 5g). However, GPR81 deficiency led to a lower expression of synaptic plasticity-related proteins (Figure 5c–g). Furthermore, a comparison of shGPR81/HIIT mice to shGPR81/SED mice revealed that HIIT failed to promote synaptic plasticity-associated protein expression (Figure 5). These results demonstrate that GPR81 can mediate HIIT-induced synaptic plasticity. Given that mitochondria are critical for synaptic plasticity, we hypothesized that the HIIT-induced upregulation of synaptic plasticity may be related to the promotion of mitochondrial function, which needs further investigation.

### 3.6. Knockdown of Hippocampal GPR81 Inhibits HIIT-Induced ERK1/2 Phosphorylation in Mice

GPR81 promotes ERK1/2 phosphorylation, and ERK1/2 can improve mitochondrial function in many tissues and cells. Therefore, we hypothesized that ERK1/2 plays an important role in GPR81-mediated hippocampal mitochondrial function and measured ERK1/2 phosphorylation in the current study. As shown in Figure 6, HIIT increased ERK1/2 phosphorylation in the hippocampi of mice, whereas GPR81 deficiency led to a reduction in ERK1/2 phosphorylation. Therefore, we hypothesized that GPR81 can regulate ERK1/2 phosphorylation in the hippocampus, which might further enhance mitochondrial function. To test this hypothesis, in vitro studies were performed.

### 3.7. Inhibition of ERK1/2 Phosphorylation Impairs L-Lactate- or GPR81 Agonist-Induced Increases in MMP in Neuro-2A Cells

To explore whether ERK1/2 could mediate lactate/GPR81-regulated mitochondrial function, Neuro-2A cells were treated with L-lactate, a GPR81 agonist (3OBA), and an ERK1/2 inhibitor (SCH) in vitro. MMP is an indicator of mitochondrial function. Therefore, we investigated whether L-lactate and 3OBA could increase MMP. The results showed that both L-lactate and 3OBA treatment led to a higher MMP (Figure 7), but this effect was blocked by SCH.

### 3.8. Inhibition of ERK1/2 Phosphorylation Impairs L-Lactate- or GPR81 Agonist-Induced Promotion of OXPHOS and ATP Production in Neuro-2A Cells

To further investigate the effect of L-lactate and GPR81 agonists on mitochondrial function, OXPHOS gene expression and ATP levels were measured in the current study. Compared to the CON conditions, L-lactate treatment led to higher gene expression of *NDUFS8* (Figure 8a) and higher ATP levels (Figure 8f). Consistent with the effect of L-lactate, the 3OBA treatment effectively upregulated the expression of genes associated with OXPHOS, including *NDUFS8* (Figure 8a), *SDHb* (Figure 8b), *Uqcrc1* (Figure 8c), and *Atp5a1* (Figure 8e). In addition, ATP levels were higher in the 3OBA group than in the DMSO group (Figure 8f). However, SCH treatment abolished the effects of L-lactate and 3OBA, which inhibited OXPHOS gene expression and reduced ATP levels, suggesting that it impaired mitochondrial function. These results demonstrate that ERK1/2 mediates the effect of lactate/GPR81 on mitochondrial function in Neuro-2A cells.

### 3.9. Inhibition of ERK1/2 Phosphorylation Impairs L-Lactate- or GPR81 Agonist-Induced Promotion of Mitochondrial Biogenesis in Neuro-2A Cells

Compared to the CON conditions, the L-lactate treatment increased the expression of genes associated with mitochondrial biogenesis, including *PGC-1α* (Figure 9a) and *NRF1* (Figure 9b), and upregulated the expression of proteins associated with mitochondrial biogenesis, including PGC-1α (Figure 9e), NRF1/2 (Figure 9f,g), and TFAM (Figure 9h). As expected, 3OBA led to the increased expression of genes and proteins associated with mitochondrial biogenesis, which was similar to the effects of L-lactate. Furthermore, the inhibition of ERK1/2 phosphorylation blocked the effects of L-lactate and GPR81 agonist on mitochondrial biogenesis, indicating that lactate/GPR81-regulated mitochondrial biogenesis is dependent on ERK1/2 signaling.

### 3.10. Inhibition of ERK1/2 Phosphorylation Impairs L-Lactate- or GPR81 Agonist-Induced Promotion of Mitochondrial Fusion in Neuro-2A Cells

Then, we measured the expression of genes and proteins related to mitochondrial fusion in Neuro-2A cells. Compared to the CON conditions, L-lactate treatment increased the expression of genes and proteins associated with mitochondrial fusion (Figure 10a,c–e). 3OBA led to the increased expression of MFN2 (Figure 10e). However, the inhibition of ERK1/2 phosphorylation blocked the effects of L-lactate and 3-OBA on mitochondrial fusion.

### 3.11. Inhibition of ERK1/2 Phosphorylation Impairs L-Lactate- or GPR81 Agonist-Induced Promotion of Synaptic Plasticity in Neuro-2A Cells

Given that mitochondria play an important role in synaptic plasticity, we next investigated whether lactate/GPR81-mediated synaptic plasticity is dependent on ERK1/2. As expected, both L-lactate and the GPR81 agonist increased the protein expression of synaptic plasticity markers, including Arc (Figure 11b), c-Fos (Figure 11c), Egr1 (Figure 11d), BDNF (Figure 11e), SYN (Figure 11f), and PSD95 (Figure 11g). However, the inhibition of ERK1/2 phosphorylation blocked the effects of L-lactate and the GPR81 agonist on the expression of synaptic plasticity proteins. Together, these results indicate that lactate/GPR81 can mediate synaptic plasticity through ERK1/2 signaling. The lactate-/GPR81-mediated promotion of mitochondrial function may contribute to synaptic plasticity.

## 4. Discussion

The present study demonstrated that hippocampal GPR81 deficiency impaired mitochondrial biogenesis, fusion, OXPHOS, synaptic plasticity and ERK1/2 phosphorylation induced by HIIT. In addition, both L-lactate and a GPR81 agonist enhanced mitochondrial biogenesis, fusion, ATP production, OXPHOS, and synaptic plasticity in Neuro-2A cells, while these effects were blocked by the inhibition of ERK1/2 phosphorylation. Collectively, these findings provide evidence that HIIT can promote mitochondrial function and synaptic plasticity in the hippocampus via the lactate/GPR81-ERK1/2 pathway (Figure 12).

Mitochondria are essential for calcium homeostasis and ATP production in the brain, processes that are vital for neuronal integrity and synaptic transmission [4,34,35]. Previous studies have shown that aerobic exercise can increase mitochondrial biogenesis, fusion, and mitochondrial function in the brain, which might further promote synaptic plasticity [36,37,38,39]. Recently, HIIT has been shown to be as effective as aerobic exercise in improving mitochondrial function. A single bout of high-intensity exercise can increase hippocampal PGC-1α mRNA levels and mitochondrial DNA (mtDNA) copy numbers [40]. In addition, one week of HIIT has been shown to increase mitochondrial content and neuroplasticity in the hippocampus of mice [5], but the underlying mechanisms remain unclear. Consistent with previous research, the current study showed that six week of HIIT increased the mitochondrial number, OXPHOS gene expression, ATP levels, mitochondrial biogenesis, mitochondrial fusion, and synaptic plasticity in the hippocampi of mice. Our previous work has shown that HIIT induces elevations in blood lactate and hippocampal lactate levels as well as mitochondrial biogenesis and fusion in the hippocampi of mice [14], which indicates that lactate is a potential participant in the HIIT-induced promotion of hippocampal mitochondrial function.

Lactate is a natural ligand of GPR81 [22], and GPR81 has been discovered to participate in regulating several metabolic processes, such as inflammation [41,42,43,44], fatty acid utilization [22,45,46,47], oblast differentiation [26], and tumor growth [24,48]. In addition, lactate/GPR81 has been demonstrated to enhance mitochondrial function in several cells, such as Müller cells [49], C2C12 cells [20], and cancer cells [50]. Moreover, a previous study has shown that GPR81 deficiency reduces mitochondrial activity and inhibits tumor cell survival [50], indicating that GPR81 is crucial for mitochondrial function. Moreover, GPR81 has been found to regulate several signaling pathways. For example, lactate/GPR81 may induce triglyceride accumulation and enhance mitochondrial function in C2C12 cells through the inhibition of the cAMP/PKA pathway [20]. Lactate increases tumor malignancy by promoting tumor small extracellular vesicle production via the GPR81/cAMP/PKA/HIF-1α axis [24]. Furthermore, the lactate/GPR81/HIF1α pathway can mediate hypoxia-induced idiopathic pulmonary fibrosis [51]. Additionally, recent works have illustrated that lactate can activate GPR81/PLC/PKC/Akt signaling during the osteoblast differentiation process [26]. Collectively, the evidence indicates that lactate/GPR81 can regulate metabolic processes and signaling pathways in different peripheral tissues and cells. Further studies should elucidate the effects of lactate/GPR81 in the brain.

In the brain, lactate/GPR81 plays important roles in energy metabolism [52], synaptic function [53], angiogenesis [54], neurogenesis, microglial activation [55], and neuronal network activity [56]. It has been shown that aerobic exercise can promote synaptic growth and prevent synaptic loss in Alzheimer’s disease, which may be related to the regulation of the GPR81/cAMP/PKA signaling pathway [25]. Previous research has also found that HIIT or lactate injection increases neurogenesis in wild-type mice but not in GPR81-knockout mice [19]. In addition, HIIT and lactate subcutaneous injection can increase blood lactate levels, brain VEGFA protein expression, and capillary density in wild-type mice but not in GPR81-knockout mice [16]. These findings suggest that lactate/GPR81 can mediate the beneficial cerebral effects of HIIT. Moreover, L-lactate administration can induce the expression of key regulators of mitochondrial biogenesis and antioxidant defense in the hippocampi of rats [57], indicating that lactate/GPR81 might regulate hippocampal mitochondrial function. Hence, in the current study, we specifically knocked down hippocampal GPR81 to identify the roles of GPR81 in the HIIT-induced promotion of mitochondrial function. We observed that hippocampal GPR81 deficiency impaired HIIT-induced mitochondrial biogenesis, fusion, OXPHOS, and synaptic plasticity, which provides compelling evidence that the HIIT-induced improvement of mitochondrial function in mice is dependent on GPR81.

As part of the downstream signal of GPR81, ERK1/2 activated by GPR81 is involved in regulating many critical metabolic processes. For example, lactate increases myotube diameter via the GRP81/MEK1/2-ERK1/2 pathway in C2C12 cells [28]. Consistent with this, GPR81 may positively affect skeletal muscle mass through the activation of the ERK1/2 pathway [27]. Most importantly, ERK1/2 can increase PGC-1α protein expression in cultured white adipocytes [58]. Additionally, a previous study has shown that the blockade of ERK1/2 phosphorylation abolishes L-lactate-induced plasticity-related gene expression in neurons [59]. Moreover, high-intensity exercise can promote cerebral angiogenesis by activating GPR81 and the ERK1/2-PI3K/Akt signaling pathways [29]. In the present study, we found that HIIT-induced mitochondrial biogenesis and fusion paralleled the increase in hippocampal ERK1/2 phosphorylation. Given the importance of ERK1/2 for mitochondrial function [30,31,32,33], we propose that many of the observed effects of lactate/GPR81-regulated hippocampal mitochondrial benefits may be related to ERK1/2 signaling.

To explore whether ERK1/2 is involved in lactate/GPR81-regulated mitochondrial function, in vitro studies were performed. A previous study has shown that lactate activates the mitochondrial electron transport chain (ETC) to increase mitochondrial ATP production and increase the cellular oxygen consumption rate (OCR) in HepG2 cells [60]. Consistent with this, our results showed that both L-lactate and a GPR81 agonist increased OXPHOS, ATP levels, and MMP in Neuro-2A cells. However, these effects were impaired by the inhibition of ERK1/2 phosphorylation. Therefore, lactate/GPR81-ERK1/2 can regulate the promotion of mitochondrial function in Neuro-2A cells. Furthermore, the change in the ATP/AMP ratio should be analyzed in the cellular model to verify the energetic cellular status in the future. Given that the promotion of mitochondrial function may contribute to synaptic plasticity, we further assessed the expression of proteins related to synaptic plasticity, which was increased in L-lactate- and GPR81 agonist-treated Neuro-2A cells. Therefore, lactate-/GPR81-mediated synaptic plasticity may be associated with the promotion of mitochondrial function in Neuro-2A cells, which needs further investigation.

## 5. Conclusions

This study demonstrates that hippocampal GPR81 deficiency impairs HIIT-induced mitochondrial biogenesis and fusion and ERK1/2 phosphorylation in the hippocampi of mice. In addition, the inhibition of ERK1/2 phosphorylation blocks the effects of L-lactate and a GPR81 agonist on mitochondrial function in Neuro-2A cells. This is the first study to show that HIIT can promote mitochondrial function and synaptic plasticity in the hippocampi of mice via the lactate/GPR81-ERK1/2 pathway. Our work reveals that the promotion of mitochondrial function contributes to synaptic plasticity, which merits further investigation.

## Figures and Tables

**Figure 1 antioxidants-12-02087-f001:**
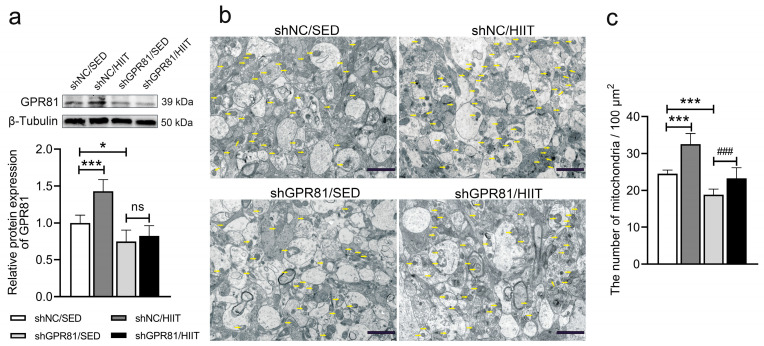
Hippocampal GPR81 deficiency decreases the number of mitochondria. (**a**) Representative Western blot image and quantification of the relative protein level of GPR81 (*n* = 6 per group). The results were normalized to the average of all the shNC/SED samples. (**b**) Representative electron micrographs of mitochondria in the CA1 region of the hippocampus. Scale bar = 2 μm. The arrowheads indicate mitochondria. (**c**) Quantification of the number of mitochondria. The data are presented as the mean ± SD. * *p* < 0.05, *** *p* < 0.001 versus the shNC/SED group. ### *p* < 0.001 versus the shGPR81/SED group. ns, no significant difference.

**Figure 2 antioxidants-12-02087-f002:**
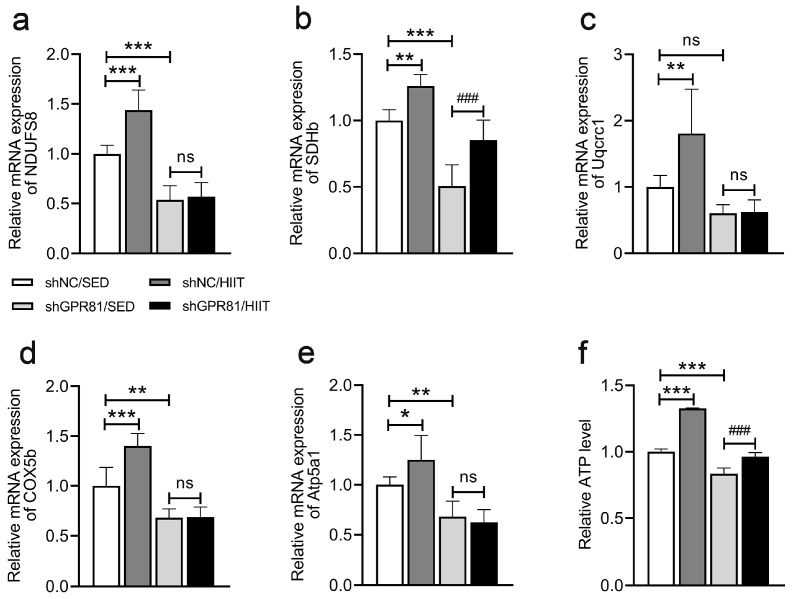
Hippocampal GPR81 deficiency decreases the expression of genes associated with OXPHOS and the levels of ATP. (**a**–**e**) Quantification of the relative mRNA levels of *NDUFS8*, *SDHb*, *Uqcrc1*, *COX5b*, and *Atp5a1* (*n* = 6 per group). (**f**) Relative ATP levels (*n* = 4 per group). The data are presented as the mean ± SD. * *p* < 0.05, ** *p* < 0.01, *** *p* < 0.001 versus the shNC/SED group. ### *p* < 0.001 versus the shGPR81/SED group. ns, no significant difference.

**Figure 3 antioxidants-12-02087-f003:**
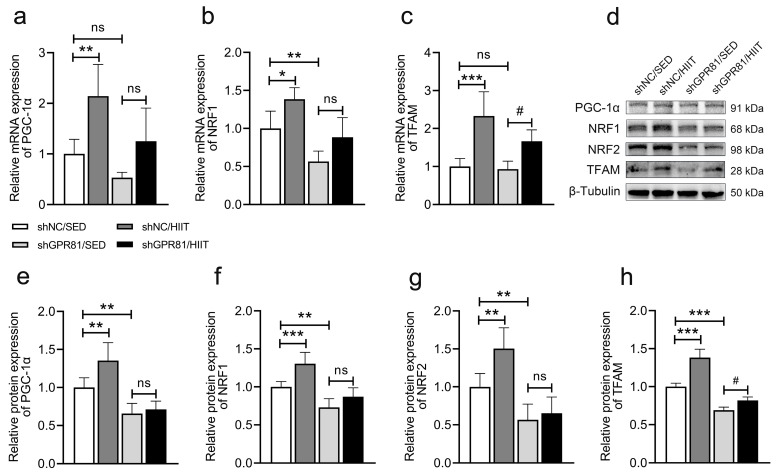
Hippocampal GPR81 deficiency impairs the expression of genes and proteins associated with mitochondrial biogenesis. (**a**–**c**) Quantification of the relative mRNA levels of *PGC-1α*, *NRF1*, and *TFAM* (*n* = 6 per group). (**d**) Representative Western blot image. (**e**–**h**) Quantification of the relative protein levels of PGC-1α, NRF1, NRF2, and TFAM (*n* = 6 per group). The data are presented as the mean ± SD. * *p* < 0.05, ** *p* < 0.01, *** *p* < 0.001 versus the shNC/SED group. # *p* < 0.05 versus the shGPR81/SED group. ns, no significant difference.

**Figure 4 antioxidants-12-02087-f004:**
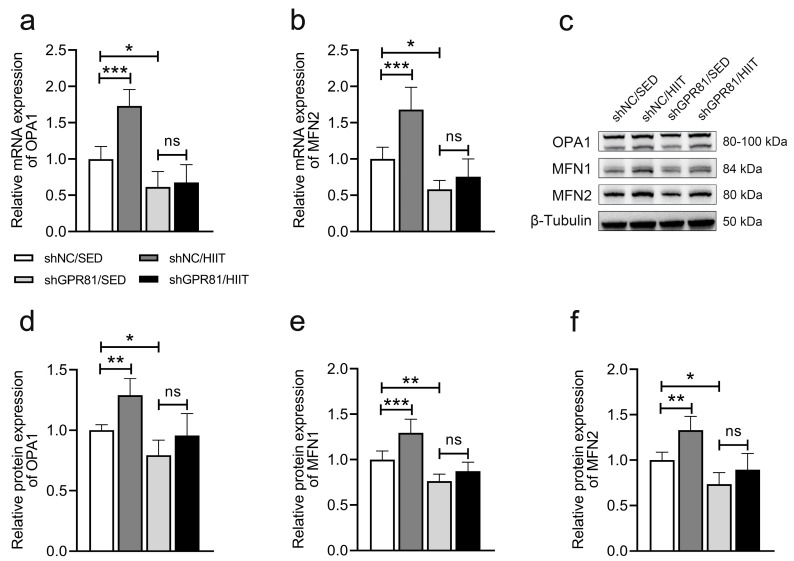
Hippocampal GPR81 deficiency impairs the expression of genes and proteins associated with mitochondrial fusion. (**a**,**b**) Quantification of the relative mRNA levels of *OPA1* and *MFN2* (*n* = 6 per group). (**c**) Representative Western blot image. (**d**–**f**) Quantification of the relative protein levels of OPA1, MFN1, and MFN2 (*n* = 6 per group). The data are presented as the mean ± SD (*n* = 6 per group). * *p* < 0.05, ** *p* < 0.01, *** *p* < 0.001 versus the shNC/SED group. ns, no significant difference.

**Figure 5 antioxidants-12-02087-f005:**
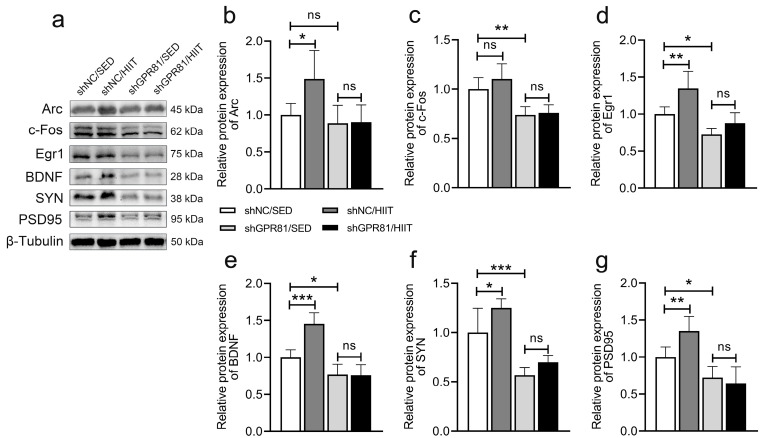
Hippocampal GPR81 deficiency impairs the expression of proteins associated with synaptic plasticity. (**a**) Representative Western blot image. (**b**–**g**) Quantification of the relative protein levels of Arc, c-Fos, Egr1, BDNF, SYN, and PSD95 (*n* = 6 per group). The data are presented as the mean ± SD. * *p* < 0.05, ** *p* < 0.01, *** *p* < 0.001 versus the shNC/SED group. ns, no significant difference.

**Figure 6 antioxidants-12-02087-f006:**
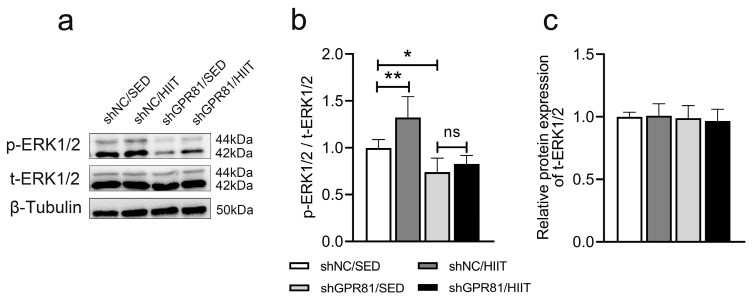
Hippocampal GPR81 deficiency inhibits ERK1/2 phosphorylation in mice. (**a**) Representative Western blot image. (**b**,**c**) Quantification of the relative protein levels of p-ERK1/2 and t-ERK1/2 (*n* = 6 per group). The data are presented as the mean ± SD. * *p* < 0.05, ** *p* < 0.01 versus the shNC/SED group. ns, no significant difference.

**Figure 7 antioxidants-12-02087-f007:**
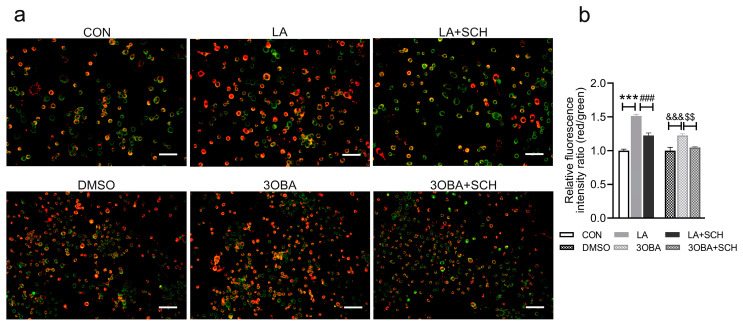
Inhibition of ERK1/2 phosphorylation impairs mitochondrial function in Neuro-2A cells. (**a**) Representative images of JC-1 staining. Scale bar = 100 μm. Red represents the JC-1 aggregate signal; green represents the JC-1 monomer signal. (**b**) Quantification of JC-1 staining (*n* = 3 per group). The data are presented as the mean ± SD. *** *p* < 0.001 versus the CON group. ### *p* < 0.001 versus the LA group. &&& *p* < 0.001 versus the DMSO group. $$ *p* < 0.01 versus the 3OBA group.

**Figure 8 antioxidants-12-02087-f008:**
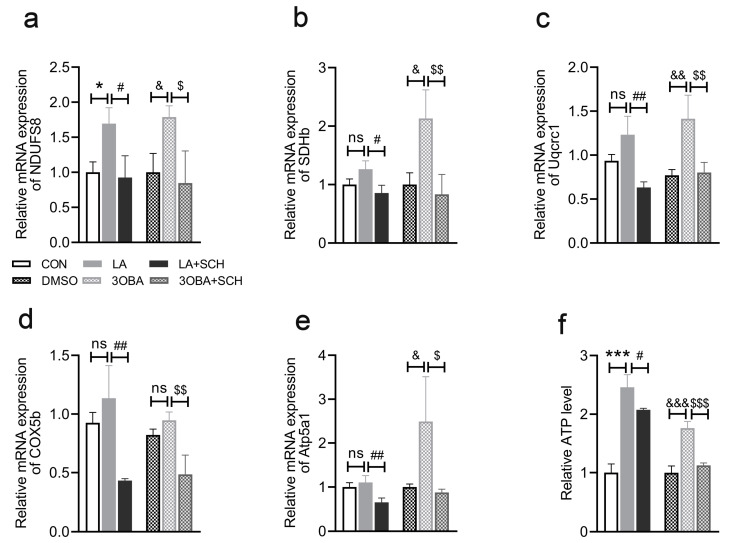
Inhibition of ERK1/2 phosphorylation impairs mitochondrial function in Neuro-2A cells. (**a**–**e**) Quantification of the relative mRNA levels of *NDUFS8*, *SDHb*, *Uqcrc1*, *COX5b*, and *Atp5a1* (*n* = 3 per group). (**f**) Relative ATP levels (*n* = 3 per group). The data are presented as the mean ± SD. * *p* < 0.05, *** *p* < 0.001 versus the CON group. # *p* < 0.05, ## *p* < 0.01 versus the LA group. & *p* < 0.05, && *p* < 0.01, &&& *p* < 0.001 versus the DMSO group. $ *p* < 0.05, $$ *p* < 0.01, $$$ *p* < 0.001 versus the 3OBA group. ns, no significant difference.

**Figure 9 antioxidants-12-02087-f009:**
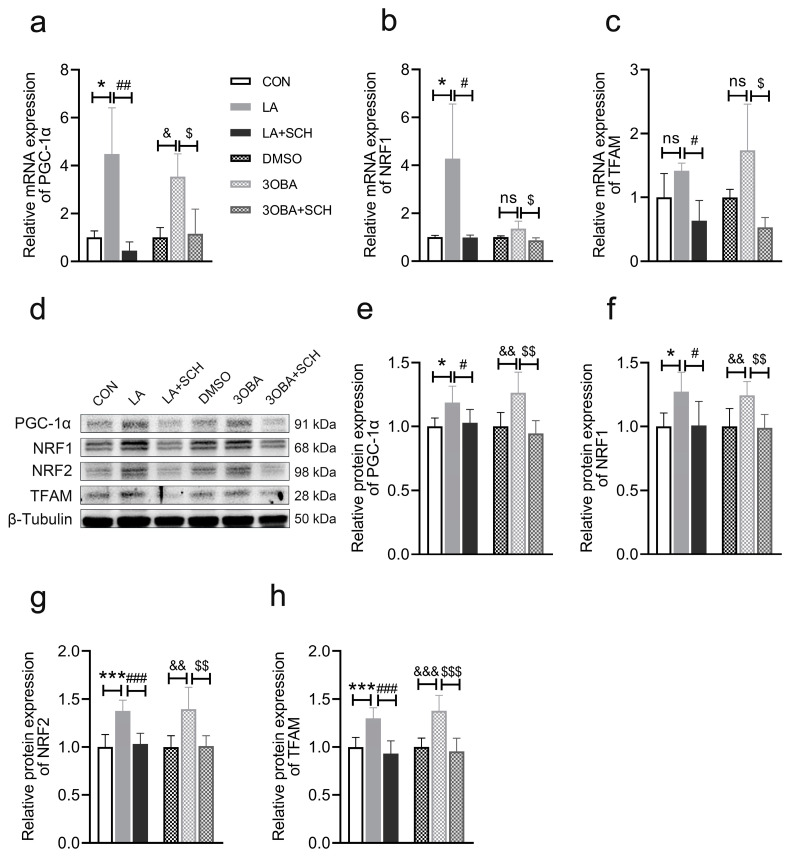
Inhibition of ERK1/2 phosphorylation impairs mitochondrial biogenesis in Neuro-2A cells. (**a**–**c**) Quantification of the relative mRNA levels of *PGC-1α*, *NRF1*, and *TFAM* (*n* = 3 per group). (**d**) Representative Western blot image. (**e**–**h**) Quantification of the relative protein levels of PGC-1α, NRF1, NRF2, and TFAM (*n* = 6 per group). The data are presented as the mean ± SD. * *p* < 0.05, *** *p* < 0.001 versus the CON group. # *p* < 0.05, ## *p* < 0.01, ### *p* < 0.001 versus the LA group. & *p* < 0.05, && *p* < 0.01, &&& *p* < 0.001 versus the DMSO group. $ *p* < 0.05, $$ *p* < 0.01, $$$ *p* < 0.001 versus the 3OBA group. ns, no significant difference.

**Figure 10 antioxidants-12-02087-f010:**
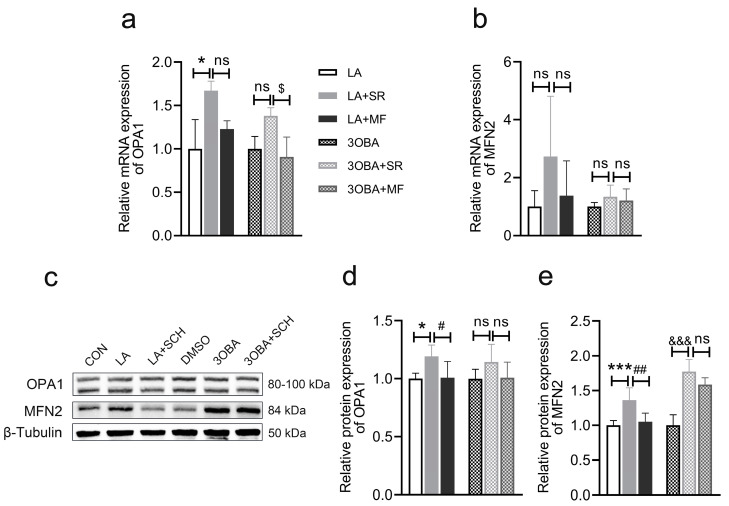
Inhibition of ERK1/2 phosphorylation impairs mitochondrial fusion in Neuro-2A cells. (**a**,**b**) Quantification of the relative mRNA levels of *OPA1* and *MFN2* (*n* = 3 per group). (**c**) Representative Western blot image. (**d**,**e**) Quantification of the relative protein levels of OPA1 and MFN2 (*n* = 6 per group). The data are presented as the mean ± SD. * *p* < 0.05, *** *p* < 0.001 versus the CON group. # *p* < 0.05, ## *p* < 0.01 versus the LA group. &&& *p* < 0.001 versus the DMSO group. $ *p* < 0.05 versus the 3OBA group. ns, no significant difference.

**Figure 11 antioxidants-12-02087-f011:**
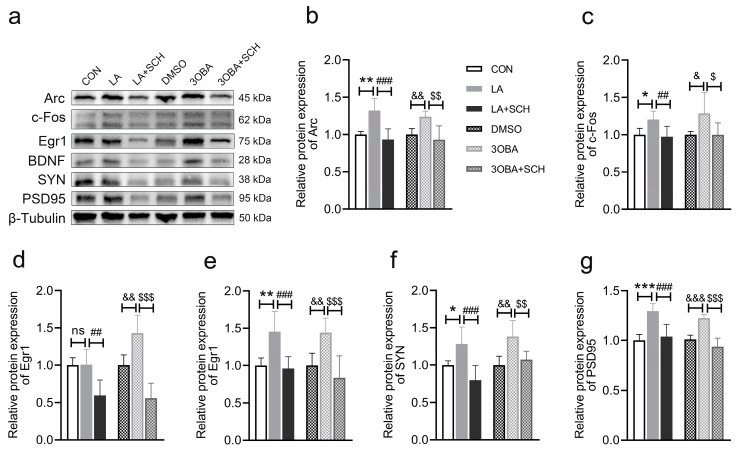
Inhibition of ERK1/2 phosphorylation impairs synaptic plasticity in Neuro-2A cells. (**a**) Representative Western blot image. (**b**–**g**) Quantification of the relative protein levels of Arc, c-Fos, Egr1, BDNF, SYN, and PSD95 (*n* = 6 per group). The data are presented as the mean ± SD. * *p* < 0.05, ** *p* < 0.01, *** *p* < 0.001 versus the CON group. ## *p* < 0.01, ### *p* < 0.001 versus the LA group. & *p* < 0.05, && *p* < 0.01, &&& *p* < 0.001 versus the DMSO group. $ *p* < 0.05, $$ *p* < 0.01, $$$ *p* < 0.001 versus the 3OBA group. ns, no significant difference.

**Figure 12 antioxidants-12-02087-f012:**
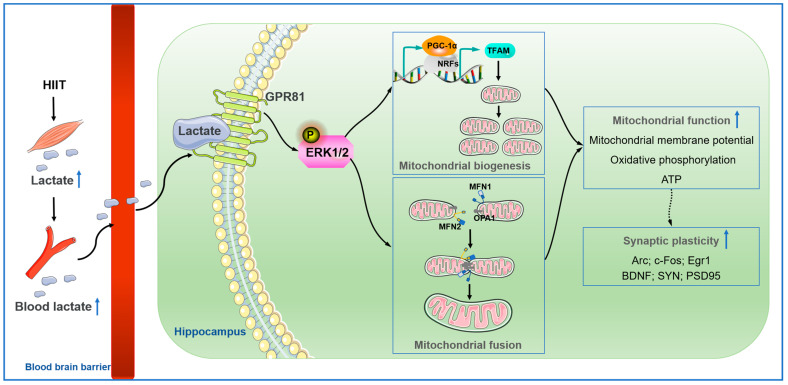
HIIT increases blood lactate levels, and lactate crosses the blood–brain barrier (BBB) into the hippocampus. Then, lactate binds the lactate receptor GPR81 and activates ERK1/2. ERK1/2 phosphorylation can promote mitochondrial biogenesis and fusion and enhance mitochondrial function, thereby increasing synaptic plasticity.

**Table 1 antioxidants-12-02087-t001:** HIIT paradigm.

Week	Warm-Up Speed (m/min)	Warm-Up Duration (min)	Training Speed (m/min)	Stimuli Duration (min)	Recovery Speed (m/min)	Recovery Duration (min)	Number of Bots	Session Duration (min)
1	10	10	22–23	4	10	2	10	60
2	10	10	22–23	4	10	2	10	60
3	10	10	26.5–28.5	4	12–14.5	2	10	60
4	10	10	26.5–28.5	4	12–14.5	2	10	60
5	10	10	30–33	4	12–15	2	10	60
6	10	10	30–33	4	12–15	2	10	60

**Table 2 antioxidants-12-02087-t002:** Sequences of the primers used in the RT-qPCR assays.

Genes	Primer	Sequence (5′-3′)	Accession Number
NDUFS8	Forward	AGTGGCGGCAACGTACAAG	NM_144870
	Reverse	TCGAAAGAGGTAACTTAGGGTCA	
SDHb	Forward	AATTTGCCATTTACCGATGGGA	NM_023374
	Reverse	AGCATCCAACACCATAGGTCC	
Uqcrc1	Forward	ACGCAAGTGCTACTTCGCA	NM_025407
	Reverse	CAGCGTCAATCCACACTCCC	
COX5b	Forward	TCTAGTCCCGTCCATCAGCAAC	NM_009942
	Reverse	GCAGCCAAAACCAGATGACAGT	
Atp5a1	Forward	TCTCCATGCCTCTAACACTCG	NM_007505
	Reverse	CCAGGTCAACAGACGTGTCAG	
PGC-1α	Forward	TTCATCTGAGTATGGAGTCGCT	NM_008904
	Reverse	GGGGGTGAAACCACTTTTGTAA	
NRF1	Forward	GTGGGACAGCAAGCGATTGTA	NM_001164230
	Reverse	TTGTACTTTCGCACCACATTCT	
TFAM	Forward	CAAAGGATGATTCGGCTCAGG	NM_009360
	Reverse	TCGACGGATGAGATCACTTCG	
OPA1	Forward	TGGAAAATGGTTCGAGAGTCAG	NM_001199177
	Reverse	CATTCCGTCTCTAGGTTAAAGCG	
MFN2	Forward	TGACCTGAATTGTGACAAGCTG	NM_133201
	Reverse	AGACTGACTGCCGTATCTGGT	
GAPDH	Forward	AGGTCGGTGTGAACGGATTTG	NM_008084
	Reverse	TGTAGACCATGTAGTTGAGGTCA	

## Data Availability

All the data and materials are available.

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
