# Peer review of "Lactate Mediates High-Intensity Interval Training—Induced Promotion of Hippocampal Mitochondrial Function through the GPR81-ERK1/2 Pathway"

_antioxidants, 2023, doi:10.3390/antiox12122087_

Round 1
Reviewer 1 Report
Comments and Suggestions for Authors
Their main finding of this study ”Lactate mediates HIIT- induced promotion of mitochondrial 2 function in the mouse hippocampus through the GPR81- 3 ERK1/2 pathway” reveal that High-intensity interval training can promote mitochondrial function and synaptic plasticity in the hippocampi of mice via the lactate/GPR81-ERK1/2 pathway, using a lactate receptor GPR81 knockout mouse model.
Moreover, the authors have performed a great number of experiments in a Neuro-2A cellular model. The design of study is well defined and based on their previous studies where their results demonstrated that hippocampal lactate levels increase significantly in mice during HIIT.
However, in order to strengthen their conclusion and main objective they should provide new data and explain clearly specific points of the discussion. Specifically, in order to reinforce the mitochondrial and ATP role in their model.
Main Remarks:
1. Title is too long. The authors should resume it. They should avoid the HITT term and include its significance.
2. In the abstract the authors should eliminate the results related to hippocampal mitochondrial fusion and biogenesis. In the present study they did not analyze it.
3. In Methods, the authors should indicate the male animal age.
4. In page 5, the authors mentioned that “The densities of the protein blots were quantified using ImageJ and normalized to those of the control group” they should explain if it was used as an endogenous protein. Did they find differences in Beta-tubulin?
5. The authors should analyze the AMP levels in the cellular model. They should discuss the obtained results (i.e ATP/AMP ratio).
6. The authors should analyze the mitochondria function by determining the electron transport chain enzymes mitochondria or perform a seahorse metabolic assay.
7. In Fig. 7a the authors should include fluorescence images in a higher magnification.
Reviewer 2 Report
Comments and Suggestions for Authors
This study tests the hypothesis that lactate produced through HIIT can activate the GPR81-ERK1/2 pathway which leads to hippocampal mitochondrial biogenesis, fusion and overall mitochondrial function ion a mouse model.
The abstract and Introduction are well written and clearly summarise the study and the background for the topic respectively. The aims of the study are clearly stated.
Line 98: Delete "by"
The Materials and Methods section is also very well written. All details of the protocol are described.
The Results section is comprehensive and all figures help the reader to follow the outcomes.
The Discussion explains the findings in context using appropriate references from previous relevant work.
Reviewer 3 Report
Comments and Suggestions for Authors
The manuscript by Shang et al. with a title ”Lactate mediates HIIT-induced promotion of mitochondrial function in the mouse hippocampus through the GPR81-ERK1/2 pathway” describes the downstream signaling of lactate released during HIIT. The author used suitable methods to answer the scientific questions. To identify the molecular mechanisms, they developed a series of elegant experiments using e.g. AAV virus to knockdown lactate receptor GRP81 in hippocampus of mice, inhibitors of ERK1/2 pathway, in vitro studies using Neuro-2A cells and comprehensive characterization of mitochondrial function, mRNA and protein expression. The manuscript is well written, easy to read and understand. The figures are of high quality. The authors included an explanatory figure In the discussion with added value for the manuscript. The manuscript can be accepted in its present form. I have only minor suggestion:
Line 125: please add the provider of Neuro-2A cells
Round 2
Reviewer 1 Report
Comments and Suggestions for Authors
Thank you to the authors for the information provided in response to my suggestions and main remarks.